# A Comparison of Feathers and Oral Swab Samples as DNA Sources for Molecular Sexing in Companion Birds

**DOI:** 10.3390/ani13030525

**Published:** 2023-02-02

**Authors:** Maria-Carmen Turcu, Anamaria Ioana Paștiu, Lucia Victoria Bel, Dana Liana Pusta

**Affiliations:** 1Department of Genetics and Hereditary Diseases, Faculty of Veterinary Medicine, University of Agricultural Sciences and Veterinary Medicine Cluj-Napoca, 400372 Cluj-Napoca, Romania; 2New Companion Animals Veterinary Clinic, Faculty of Veterinary Medicine, University of Agricultural Sciences and Veterinary Medicine Cluj-Napoca, 400372 Cluj-Napoca, Romania

**Keywords:** pet birds, genetic sexing, PCR, feathers, oral swabs, DNA

## Abstract

**Simple Summary:**

Most bird species have no distinct sexual dimorphic traits. Molecular genetic sexing is considered one of the most accurate methods for sex determination in monomorphic birds. The aim of this study was to compare the sexing results obtained by PCR method from paired samples of feathers and oral swabs collected from the same individuals. Oral swabs proved to be a more reliable sample for genetic sex determination in the species tested in this study, especially for featherless, newly hatched chicks.

**Abstract:**

The early age determinism of the sex in case of monomorphic birds is very important, because most companion birds have no distinct sexual dimorphic traits. Molecular genetic sexing was proved to be one of the most accurate sex determinations in monomorphic birds. The aim of this study was to compare the results obtained by PCR performed on isolate genomic DNA from paired samples of feathers and oral swabs collected from the same individuals. Samples of oral swabs (*n* = 101) and feathers (*n* = 74) were collected from 101 companion birds from four different species (*Columba livia domestica*, *Psittacula krameri*, *Neophema splendida* and *Agapornis* spp.). The PCR was performed for the amplification of the CHD1W and CHD1Z genes in females and the CHD1Z gene in males. The overall PCR success rate of sex determination was significantly higher from oral swabs than from feathers. The PCR success rate from oral swabs was higher in juveniles and from feathers was significantly higher in adults. The similarity between the oral swab and feathers was obtained in 78.38% of the birds. Oral swabs proved to be a more reliable sample for genetic sex determination in the species tested in this study.

## 1. Introduction

The most common pet birds are the *Passeriformes*, represented mainly by canaries and finches, the *Psittaciformes*, also known as parrots, represented by roughly 400 species, and the *Columbiformes*, commonly known as pigeons or doves [1,2]. More than 50% of the worldwide population of birds have no distinct sexual dimorphic traits [3]. Most parrots, such as Lovebirds (*Agapornis* spp.), are monomorphic, while others are sexually dimorphic when they reach sexual maturity [2]. For example, the Scarlet-chested Parrot (*Neophema splendida*) adult male is red chested, and the Rose-ringed Parakeet (*Psittacula krameri*) adult male has a dark ring drawn around his neck. However, these sexual dimorphic traits appear later in life, after the birds reach sexual maturity [2,3]. Domestic Pigeons (*Columba livia domestica*) are also monomorphic and are considered monogamous [4]. The current study includes exclusively monomorphic companion birds.

Pet birds are highly social and need pairing. Therefore, molecular sexing helps provide early appropriate welfare facilities for birds [5]. The determination of sex in parrots can be performed through traditional methods, surgical methods and genetic methods [3,6,7]. Surgical and traditional methods are time consuming, costly, and even life threatening, and have low sensitivity [8,9]. Genetic sexing presents multiple advantages such as accuracy and precision in sexing. Moreover, it is considered a safer method because sample collection only subjects the birds to a few minutes of handling stress and does not put them at risk of infection, compared with endoscopic sexing. The main advantage is that any bird, regardless of size or age, can be safely sexed, particularly newly hatched chicks [8,9,10,11]. Genetic sexing of birds has many applications in different domains such as behavioral medicine, conservative medicine, wildlife management, breeding of different species of birds in evolutionary studies or in forensic medicine. Moreover, molecular sexing helps improve the breeding programs of birds in captivity and facilitates poultry reproduction strategies [10,12,13,14].

Sex chromosomes in birds (ZW) are different from mammals (XY). Male birds have two identical sex chromosomes (ZZ), whereas females are heterogametic (ZW). Sex identification is performed by the amplification of the CHD gene (chromo-helicase-DNA binding protein gene) found on the sex chromosomes of birds. Females have the CHD1W and CHD1Z genes, while males only have the CHD1Z gene [6,15]. PCR-based sexing is one of the most reliable sexing methods applied to birds but, nonetheless, laboratory protocols often require optimization for each species [16]. 

Biological samples suitable for genetic sexing in birds are divided into invasive (blood sample, biopsy), moderately invasive (plucked feathers, oral swabs) and non-invasive (molted feathers, eggshell membranes, feces; and from carcasses: blood, liver, brain, gonads, intestine, kidneys, skin, bone, tissues in various stages of autolysis) [17,18,19]. The PCR success rate using the most common tissues sampled from birds is described in Table 1.

Moderately invasive collected samples (feather and oral swabs) are recommended for sexing extremely young chicks, adult birds of small sizes or any bird for which blood sampling is considered to be too invasive [16,17,24]. Feather sampling provides lower DNA yields but is considered faster and easier than blood sampling. Moreover, feather sampling is considered safer, because the reduced handling time may decrease stress during capture [16]. Buccal epithelial cells can be sampled through buccal/oral swabs [24]. Although oral swabs have been used for molecular bird sexing before [24], only a few studies evaluated the reliability of oral swabs for sex determination in birds, especially in very young nestlings [20,21,22,23]. Molecular sexing using blood samples has yielded more accurate because the blood samples contain a higher DNA concentration than oral swabs or feather samples. However, bird handling for feather and oral swab collection is faster, moderately invasive, and storage conditions are much easier than in the case of blood samples [16,24,25]. Nevertheless, there are studies that recommend blood sampling in favor of feather plucking or clipping, particularly for wild birds [26]. 

The aim of this study was to compare the results obtained by PCR performed on isolate genomic DNA from paired samples of feathers (processed using TissueLyser II (Qiagen, Hilden, Germany) in order to increase the amount of DNA) and oral swabs, collected from the same individuals. It also aimed to find an alternative to invasive sampling (blood collection) for sexing newly hatched birds. The present study offers oral swab sampling as a moderately invasive solution for sexing featherless, newly hatched chicks. According to our knowledge, the current study reports, for the first time, feather versus oral swab-based sex determinations in the following species: *Psittacula krameri*, *Neophema splendida* and *Agapornis* spp. Pairing parrots has been proven to increase welfare, therefore, early sex determination can present great value for bird owners.

## 2. Materials and Methods

### 2.1. Sample Collection

From December 2021 to March 2022, samples of oral swabs and feathers were randomly collected from 101 companion birds from the orders *Columbiformes* and *Psittaciformes* (*Columba livia domestica*, *Psittacula krameri*, *Neophema splendida* and *Agapornis* spp.) in Romania. The study included 31 adult birds and 70 juveniles [2,4], of which 80 were live birds belonging to all four species tested, and 21 cadavers belonging to *Columba livia domestica*. From these 101 birds, 74 paired samples of feather-oral swabs from the same individuals and 27 additional oral swabs from featherless, newly hatched birds were collected [2,4] (Table 2). Additionally, from the 21 deceased *Columba livia domestica,* blood clots were collected (Table 2). All of the samples were collected during the routine check-ups of companion live birds or cadavers admitted to the New Companion Animals veterinary clinic of the Faculty of Veterinary Medicine, University of Agricultural Sciences and Veterinary Medicine, Cluj-Napoca, Romania. All of the individuals included in the current study showed no distinct sexual dimorphic traits. All data regarding the birds’ age was provided by the owners or breeders. The owners gave consent for these procedures. 

Contour feathers (2 to 4) with intact calamuses were sampled from the wings or abdominal region and then stored in plastic bags. Chick down feathers were successfully collected from a seven day old *Columba livia domestica*. In featherless, newly hatched (*n* = 27) Rose-ringed Parakeets (*Psittacula krameri*), feathers could not be sampled due to a lack of feather growth, but oral swabs were easily collected (Figure 1). 

Oral swabs were collected using sterile cotton swabs (Prima, Taizhou Honod Medical Co., Ltd., Zhejiang, China), according to the protocol described by Handel et al. [24] (Figure 1). All samples were collected using surgical gloves, labeled individually and stored at −20 °C until processing.

### 2.2. DNA Extraction and Polymerase Chain Reaction (PCR)

From April 2022 to September 2022, DNA samples (oral swabs, feathers and blood clots) were processed. Genomic DNA was extracted from all samples collected from the birds. For all types of tissues, the same protocol was used. DNA extraction was performed using a commercial kit (Isolate II Genomic DNA kit; Meridian Bioscience, Newtown, OH, USA) following the manufacturer’s protocol. Oral swabs were transferred to 1.5 mL Eppendorf tubes using sterile scissors. With the help of a sterile scalpel, the feather’s calamus was sectioned in small pieces of about 2–3 mm. Feathers were subjected to mechanical destruction by high-speed shaking with steel beads using TissueLyser II (Qiagen, Hilden, Germany). DNA was extracted from 25 mg of feather/blood clots from the entire swab, and further tested for the presence of specific genes CHD1W and CHD1Z by standard PCR. Amplification of the CHD gene was performed following the protocol described by Griffiths et al. [6], using P2 (5′-TCT GCA TCG CTA AAT CCT TT-3′) and P8 (5′-CTC CCA AGG ATG AGR AAY TG-3′) primers (Generi-Biotech, Hradec Králove, Czech Republic). PCR was carried out in a 25 μL reaction mixture consisting of 12.5 μL of MyTaq Red HS Mix (Meridian Bioscience, Newtown, OH, USA) and 25 pM of each primer. The volume of DNA template was 4 μL. The amplification was performed in Bio-Rad C1000TM Thermal Cycler (Bio-Rad Laboratories, Hercules, CA, USA). Cycling conditions were 95 °C for 1 min, followed by 95 °C for 15 s, 48 °C for 45 s, 72 °C for 45 s, 94 °C for 30 s (35 cycles); 48 °C for 1 min and 72 °C for 5 min. Aliquots of each PCR product were electrophoresed on 3% agarose gel stained with RedSafe Nucleic Acid Staining Solution 20,000× (iNtRON Biotechnology, Inc., Gyeonggi-do, Republic of Korea) and examined under UV light (Bio-Rad BioDoc-ItTM Imagine System, Bio-Rad Laboratories, Hercules, CA, USA). The fragment size of the DNA was compared with a 100 bp DNA ladder (Fermentas; Thermo Fisher Scientific, Waltham, MA, USA) and assigned sex by counting the visible bands in each lane. Females are characterized by obtaining two bands corresponding to the CHD1W and CHD1Z genes, while males present only one band corresponding to the CHD1Z gene.

### 2.3. Statistical Analysis

Point estimates and 95% confidence intervals (95% CI) for the PCR success rate of sex determination by each type of tissue samples (oral swab; feathers), by bird species (*Columba livia domestica*; *Psittacula krameria*; *Neophema splendida*; *Agapornis* spp.) and by age (featherless newly hatched, <3 weeks; juvenile, <6 months; adult, >6 months) were analyzed. The difference in prevalence among groups was statistically analyzed using a Chi-square test of independence. A *p*-value of <0.05 was considered statistically significant. Data were processed using EpiInfo 2000 software (CDC, Atlanta, GA, USA) (http://wwwn.cdc.gov/epiinfo, accessed on 15 November 2022).

## 3. Results

Conventional PCR with P2 and P8 primers allowed sex determination in all four tested bird species. The results of molecular sexing of birds from oral swabs and feathers are presented in Table 3. Two birds identified as male only in one type of sample (oral swab or feathers) were considered as unidentified unless the result could be confirmed by another independent analysis (PCR on blood clots).

The overall PCR success rate of sex determination was significantly higher from oral swabs (94.06%; 95/101; CI 95%: 87.52–97.79; *χ*^2^ = 4.7426, *df* = 1, *p* = 0.0294) than from feathers (82.43%; 61/74; CI 95%: 71.83–90.30). 

From oral swabs, the PCR success rate was higher in juveniles (94.29%; 66/70; CI 95%: 86.01–98.42; *χ*^2^ = 0.0209, *df* = 1, *p* = 0.8850) than in adult birds (93.55%; 29/31; CI 95%: 78.58–99.21). 

From feathers, the PCR success rate was significantly higher in adults (96.77%; 30/31; CI 95%: 83.30–99.92; *χ*^2^ = 7.5774, *df* = 1, *p* = 0.0059) than in juvenile birds (72.09%; 31/43; CI 95%: 56.33–84.67).

A similarity between sex determination from the oral swab versus feather samples was obtained in 58/74 birds (78.38%; CI 95%: 67.73–86.23). 

In Domestic Pigeons, the same results from oral swabs and from feathers were obtained in 38/43 birds (88.37%; CI 95%: 75.52–94.93), five samples from different deceased pigeons were unidentified (three from oral swabs collected from deceased pigeons that had lesions in the oral cavity and two from feathers). All of the 21 blood clot samples collected from deceased Domestic Pigeons were successfully used for sex identification, and the results were identical with those obtained from oral swabs or feathers. 

In Rose-ringed Parakeets, the similarity between oral swab and feathers was obtained in 11/15 (73.33%; CI 95%: 48.05–89.10), in Lovebirds 7/8 (87.50%; CI 95%: 52.91–97.76) and in Scarlet-chested Parrot 2/8 (25%; CI 95%: 7.15–59.07). One individual of Rose-ringed Parakeets and one Scarlet-chested Parrot identified as a male only in oral swab samples were considered unidentified. In one individual of the Lovebirds, the sex could not be identified because neither the oral swab nor the feather samples could be amplified.

The PCR products showed two bands in females corresponding to the CHD1W and CHD1Z genes and a single band in male corresponding to the CHD1Z gene (Figure 2 and Figure 3, original PCR gel electrophoresis figures were shown in Appendix A), which range in size between 350 bp and 450 bp.

## 4. Discussion

The present study showed a PCR success rate of sex determination from feathers samples of 82.43%. Intact contour feathers (2 to 4) of all sizes were plucked or clipped from companion birds. In newly hatched Rose-ringed Parakeets, feathers could not be sampled due to a lack of feather growth. Both types of feather samples, plucked and clipped, collected from Domestic pigeons provided similar DNA templates for the molecular sexing reactions. From feather samples collected from two juvenile Domestic Pigeons, four juvenile Rose-ringed Parakeets, six juvenile Scarlet-chested Parrots and one adult Lovebird, no amplicons were obtained. These individuals presented small feather samples, but oral swabs yielded a good amount of DNA for sex determination. Similar to previous studies [18,21], DNA amplification using plucked feather samples could not be performed in all of the birds tested in this study. The PCR success rate of sex determination using plucked feathers ranged from 60% [18] to 97–100% [21].

The overall PCR success rate of sex determination from the feathers samples included in the present study was higher in adult birds (96.77%) than in juveniles (72.09%). Contrariwise, in this study, reliable results were obtained from down samples in the case of a 7-day-old Domestic Pigeon. Plucked chick down has a lower DNA concentration than feathers but had a 95.9% success rate in sex determination of Greater Sage-grouse (*Centrocercus urophasianus*) [17]. 

The decrease in the accuracy of molecular sexing when using feather samples might be caused by sampling contour feathers, which are small in size with poorly developed calamus and show a low amount of DNA [18]. Moreover, higher quality DNA is provided by freshly plucked feathers than by molted feathers [16,18]. Even though shed feathers are normally considered inferior samples, in some cases they might result from bird fighting and can be equivalent in quality to plucked feathers [17]. No molted feathers were processed in the current study. Shed feathers proved to have an overall PCR sex identification success rate below 50% [18,23]. It is recommended to collect double the quantity of shed feather samples as the needed for the sample size in order to carefully plan the sampling strategy and analysis [18]. Shed feathers could remain in the field for several months before amplification, therefore, environmental events can compromise DNA quality leading to variable results and unpredictable errors compared with plucked feathers or oral swabs [21]. The feather preservation method can have negative effects on DNA quantity and quality. Samples should be stored in a freezer with a quick-freeze function, preventing the formation of ice crystals that can degrade DNA [16].

In order to obtain an increased concentration of DNA from feather samples, the preliminary lysis of the cells was performed by mechanical shock with the help of TissueLyser II (Qiagen, Hilden, Germany) with stainless steel beads [25]. Despite this, the PCR success rate of sex-determination from feathers was significantly lower than that from oral swabs (94.06%; *p* = 0.0294). Moreover, for higher DNA quantity, two to four contour feathers with intact calamuses sampled from the wings or abdominal region of birds were processed for each individual. 

Oral swab samples were previously tested for molecular sexing in different companion bird species and are considered a good option for sexing monomorphic young companion birds [25]. Various PCR amplification success rates from oral swabs were reported: 100% [21], 91% [22], 87% [23] and ranging between 71.4 to 85.7% [20]. The PCR success rate of sex determination from oral swab samples included in the present study was 94.06%, being higher in juvenile birds (94.29%) than in adult birds (93.55%). Similar results were obtained from oral swabs sampled from Ivory Gulls (*Pagophila eburnea*), 100% in juveniles and 85% in adults [23].

The current study included both live companion birds (*n* = 80) and deceased (*n* = 21) Domestic Pigeons. Oral swab and feather samples collected from deceased Domestic Pigeons in various degrees of decomposition provided adequate DNA templates for the molecular sexing reactions, with the exception of the oral swabs collected from three pigeons with oral pseudo membranes. However, there are studies that claimed that using biological samples collected from birds in advanced stages of decomposition does not achieve reproducible results [27] because highly decomposed corpses contain a lower number of cells and DNA compared to fresh cadavers [17,19,28,29,30]. There are no previous studies regarding sampling oral epithelial cells from birds with mouth lesions in order to determine their sex. We assume that these inconsistent results appeared due to the reduced number of epithelial cells caused by oral cavity lesions, added to the advanced state of decay of the corpses. All of the blood clots samples collected from the deceased Domestic Pigeons were successfully used for sex identification, and the results were identical to those obtained from oral swabs or feathers.

In some cases, the female-specific W-chromosome PCR band was consistently weaker than the Z-chromosome band, therefore, the weaker amplification of the diagnostic band could result in females being misdiagnosed as males [16]. In such cases, samples from females would show only a single band (from the W-chromosome), that could result in them being sexed as males, while samples from males would show absolutely no band [16,31]. For this reason, in the present study, all of the birds identified as male in only one sample type (oral swabs or feathers) were considered as unidentified. The PCR female results are definitely reliable (two bands Z and W-chromosomes), while PCR male results (Z-chromosome) cannot be guaranteed, as they could actually be a false negative for a female. On the other hand, the myth of the universal sexing marker has been debunked for *Psittaciformes* because of the different patterns of the PCR products depending on the tested species [31]. 

Aiming for tissues that can be sampled by moderately invasive methods, both feather and oral swab samples are recommended. Feather versus oral swab-based sex determinations were previously studied [17,21,25,27,32], but not in the species mentioned in the current study. Various feather sampling methods (plucked versus clipped) have also been described in corticosterone measurement studies in male and female wild birds [33,34] and proved that feather plucking does not increase corticosterone levels in wild birds [35]. Recently, oral swabs have started to be used for salivary corticosterone measurements in wild birds [36].

Oral swabs present greater ease in sampling than plucked feathers. The bird handling time is short, therefore, stress might be reduced, no dangerous tools/materials are used, birds are not at risk of infection and storage conditions are advantageous [32]. Moreover, oral swabs presented a greater ease in laboratory processing compared with feathers. Other authors drew similar conclusions regarding oral swab processing [25,32]. Oral swabs proved to be a more reliable sample for genetic sex determination in the species tested in this study.

If a decade ago commercial kits for DNA isolation could not effectively deal with inhibitors of the PCR reaction (keratin in feathers, hemoglobin in blood, etc.) and a significant number of failed reactions were recorded, with the advancement of technology and the appearance of new modern kits of nucleic acid isolation, this problem has been overcome. However, even if this problem has been solved, genetic sex determination still has limitations such as the size of feather samples, the degree of feather damage with or without intact calamuses, the anatomical region from where the feathers have been plucked, the number of cells collected on oral swabs, lesions in the oral cavity, etc.

## 5. Conclusions

The present study includes a comparison between paired samples of feather-oral swabs in *Psittacula krameri*, *Neophema splendida*, *Agapornis* spp. and *Columba livia domestica.* The oral swab samples proved to be more efficient compared to the processed (TissueLyser II, Hilden, Germany) feather samples and provided adequate DNA templates for the molecular sexing of juvenile and adult companion birds. Oral swab samples were exclusively used for newly hatched, featherless parrots. The current study recorded a limitation of oral swab samples in the case of the presence of the pseudo-membrane deposits in the oral cavity. Further improvements of PCR protocols for noninvasive and moderately invasive sampling are needed in order to eliminate invasive methods in birds.

## Figures and Tables

**Figure 1 animals-13-00525-f001:**
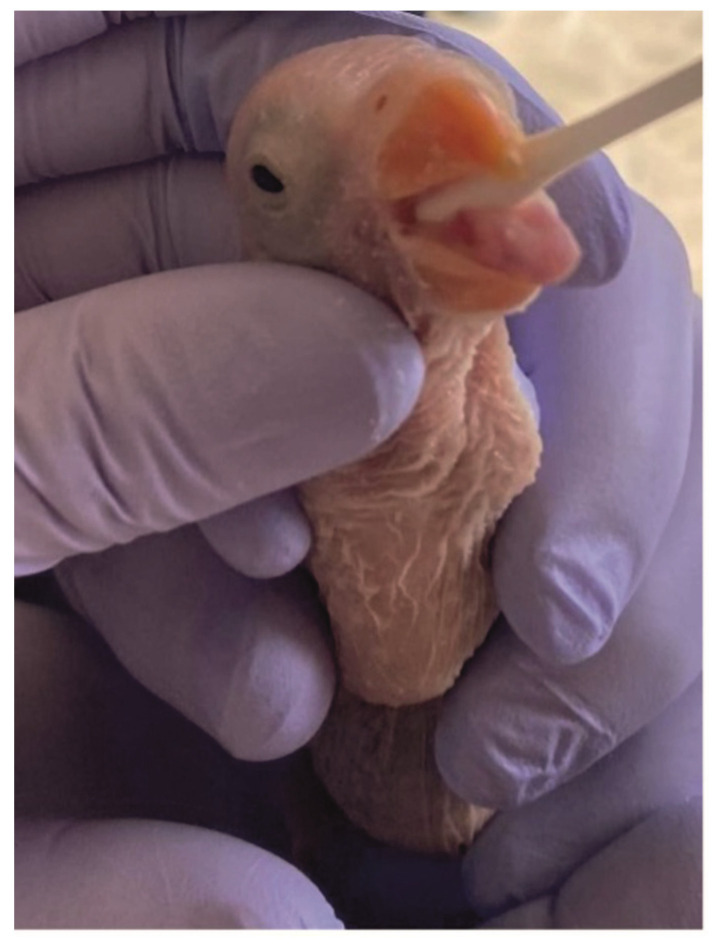
Oral swab sampling in a 7 day old Rose-ringed Parakeet. (*Psittacula krameri*).

**Figure 2 animals-13-00525-f002:**
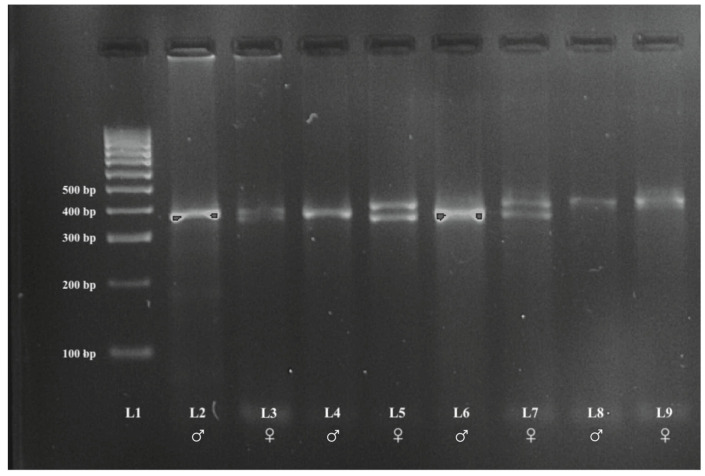
PCR results from oral swabs samples. Legend: L1—Size standard (100-bp DNA ladder), ♂—male, ♀—female, L2 and L3 Domestic Pigeons (*Columba livia domestica*), L4 and L5 Scarlet-chested Parrots *(Neophema splendida*), L6 and L7 Rose-ringed Parakeets (*Psittacula krameri*), L8 and L9 Lovebirds (*Agapornis* spp.).

**Figure 3 animals-13-00525-f003:**
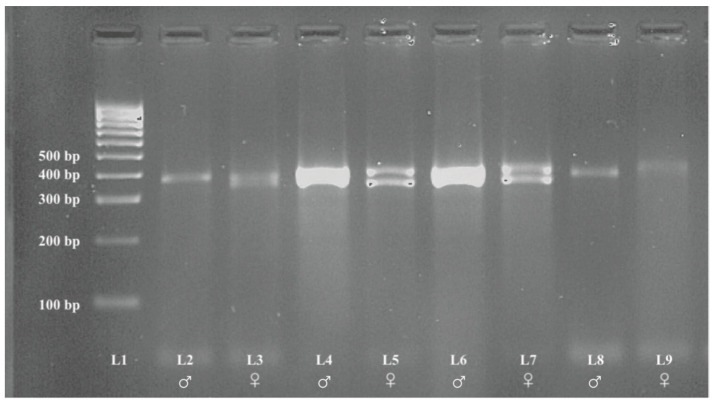
PCR results from feather samples. Legend: L1—Size standard (100-bp DNA ladder), ♂—male, ♀—female, L2 and L3 Domestic Pigeons (*Columba livia domestica*), L4 and L5 Scarlet-chested Parrots *(Neophema splendida*), L6 and L7 Rose-ringed Parakeets (*Psittacula krameri*), L8 and L9 Lovebirds (*Agapornis* spp.).

**Table 1 animals-13-00525-t001:** PCR success rate for molecular sexing in birds using different biological samples.

Sample Type	PCR Success Rate *	Bird Species	Reference
Blood	100%	12 wild bird species from the orders of *Ciconiiformes* and *Passeriformes*	[20]
	100%	Black-capped Chickadees (*Poecile atricapilla*)	[16]
Oral swabs	100%	Ivory Gull (*Pagophila eburnea*)	[21]
	91%	Common Swifts (*Apus apus*)	[22]
	87% overall 100%(juv.); 85%(adults)	Ivory Gull (*Pagophila eburnea*)	[23]
	71.4–85.7%	12 wild bird species from the orders of *Ciconiiformes* and *Passeriformes*	[20]
Plucked feathers	97–100%	Ivory Gull (*Pagophila eburnea*)	[21]
	60%	Unknown species	[18]
Moulted feathers	77–99%	Ivory Gull (*Pagophila eburnea*)	[21]
	<50%	Unknown species	[18]
	<50%	Ivory Gull (*Pagophila eburnea*)	[23]
Down feathers	95.9%	Greater Sage-grouse (*Centrocercus urophasianus*)	[17]

* PCR success rate (percentage of samples whose sex was successfully identified) for molecular sexing in birds.

**Table 2 animals-13-00525-t002:** Oral swabs and feathers collected from pet birds from five different species.

Species	Age	Sexual Dimorphism	No. of Individuals	Oral Swabs	Feathers	Blood Clots
*Columbiformes* (*n* = 43)						
Domestic Pigeons (*n* = 43) (*Columba livia domestica*)	juvenile (<6 months) [4]	monomorphic	22	22	22	6
	adult (>6 months) [4]		21	21	21	15
*Psittaciformes* (*n* = 58)						
Rose-ringed Parakeets (*n* = 42) (*Psittacula krameri*)	featherless newly hatched (<3 weeks) [2]	monomorphic (dimorphic only after sexual maturity)	27	27	-	-
	juvenile (<6 months) [2]		15	15	15	-
Lovebirds (*n* = 8) (*Agapornis* spp.)	adult (>6 months) [2]	monomorphic	8	8	8	-
Scarlet-chested Parrots (*n* = 8) (*Neophema splendida*)	juvenile (<6 months) [2]	dimorphic after sexual maturity	6	6	6	-
	adult (>6 months) [2]		2	2	2	-
TOTAL			101	101	74	21

**Table 3 animals-13-00525-t003:** Results of molecular sexing juvenile and adult pet birds included in this study.

Species	Age	Oral Swab					Feathers				
		Total	M	F	U	PCR Success Rate (%)	Total	M	F	U	PCR Success Rate (%)
Domestic Pigeons (*Columba livia*)	Juvenile	22	6	14	2	90.9	22	6	14	2	90.9
	Adult	21	12	8	1	95.24	21	13	8	-	100
Rose-ringed Parakeets (*Psittacula krameri*)	Newly hatched	27	12	15	-	100	*	*	*	*	*
	Juvenile	15	8	6	1 **	93.33	15	8	3	4	73.33
Lovebirds (*Agapornis* spp.)	Adult	8	5	2	1	87.5	8	5	2	1	87.5
Scarlet-chested Parrots *(Neophema splendida*)	Juvenile	6	0	5	1 **	83.33	6	0	0	6	0
	Adult	2	1	1	-	100	2	1	1	-	100
TOTAL		101	44	51	6	94.06	74	33	28	13	82.43

Legend: M—male; F—female; U—unidentified; PCR Success rate—percentage of samples whose sex was successfully identified using oral swabs and feather samples; *—not applicable; **—birds identified as male only in one type of samples (oral swab).

## Data Availability

All the results of the study are presented within the manuscript and its Appendix A.

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
