# Peer review of "A Comparison of Feathers and Oral Swab Samples as DNA Sources for Molecular Sexing in Companion Birds"

_animals, 2023, doi:10.3390/ani13030525_

Round 1
Reviewer 1 Report
This paper reports a comparison of feather and oral swab samples as DNA sources for molecular sexing in companion birds. The paper is mostly well written, sequenced-focused and provides good results. Some extra testing and clarifications are recommended.
Line 49. It should say "low sensitivity". I know Bermudez-Humaran et al 2008 explicitly says "sensibility", however I invite authors to review definitions on sensitivity and specificity when describing a diagnostic test. Please add a second citation supporting this.
Lines 108-109. Unclear, please rewrite. Its recommended to start with "A total of 62 samples were collected from adult birds..."
Lines 114-115. Please rewrite. Perhaps replace "using sterile pharyngeal exudate collectors" for "buccal swabs" followed by brand and manufacturer of such buccal swabs as per Handel et al 2006.
Lines 184-185. On Table 3, please change "Success rate" for "Identification rate". As mentioned by the authors, Harvey et al 2006 points out the possibility of some females being identified as males due to poor amplification of diagnostic band. As the authors did not use a gold test (such as surgically observing the gonads, or blood PCR testing), a 100% sex identification confidence cannot be claimed. Please comment on this fact under Discussion.
Lines 186-196. Discuss/mention expected amplicon sizes.
Lines 207-280. Please comment on the following:
- I did not see that DNA quantities were measured. Perhaps this might have contributed to oral swabs having an statistical difference over feathers. If possible, please, test DNA using a nanodrop or similar technology. This can shed light on the differences between these samples. This will enrich the paper and make it more citable. In any case, please comment on the lack of this data.
- No comment is done over on Harvey et al 2006 findings, that some females can be scored as males by PCR. Please comment and explore this possibility. Add proper citations.
Lines 221-226. If possible, please comment on the presence of feather pulp on feathers collected.
Author Response
Reviewer #1
This paper reports a comparison of feather and oral swab samples as DNA sources for molecular sexing in companion birds. The paper is mostly well written, sequenced-focused and provides good results. Some extra testing and clarifications are recommended.
Line 49. It should say "low sensitivity". I know Bermudez-Humaran et al 2008 explicitly says "sensibility", however I invite authors to review definitions on sensitivity and specificity when describing a diagnostic test. Please add a second citation supporting this.
AU: We have changed in text: “sensitivity [8,9]” and added a second citation.
Lines 108-109. Unclear, please rewrite. Its recommended to start with "A total of 62 samples were collected from adult birds..."
AU: To clarify, we have changed in MM: “From December 2021 to March 2022 samples of oral swabs and feathers were randomly collected from 101 companion birds from the orders Columbiformes and Psittaciformes (Columba livia domestica, Psittacula krameri, Neophema splendida and Agapornis spp.) in Romania. The study included 31 adult birds and 70 juveniles [2,4], of which 80 were live birds belonging to all four species tested and 21 cadavers belonging to Columba livia domestica. From these 101 birds, 74 paired samples of feathers-oral swabs from the same individuals and 27 additional oral swabs from featherless newly hatched birds were collected [2,4] (Table 2).”
Lines 114-115. Please rewrite. Perhaps replace "using sterile pharyngeal exudate collectors" for "buccal swabs" followed by brand and manufacturer of such buccal swabs as per Handel et al 2006.
AU: Done. We have changed in the MM: “using sterile cotton swabs (Prima, Taizhou Honod Medical Co., Ltd., Zhejiang, China) according to the protocol described by Handel et al.”.
Lines 184-185. On Table 3, please change "Success rate" for "Identification rate". As mentioned by the authors, Harvey et al 2006 points out the possibility of some females being identified as males due to poor amplification of diagnostic band. As the authors did not use a gold test (such as surgically observing the gonads, or blood PCR testing), a 100% sex identification confidence cannot be claimed. Please comment on this fact under Discussion.
AU: We have changed in text “Success rate” with “PCR success rate”.
We have explained in the legend of Table 3 what PCR success rate means: “Legend: M – male; F – female; U – unidentified; PCR Success rate - percentage of samples whose sex was successfully identified using oral swabs and feather samples; * - not applicable; ** - birds identified as male only in one type of samples (oral swab).”
There are many authors that have used “success rate/PCR success rate” with the same purpose as we [16,18,21,22,23,24].
AU: From the 21 deceased Domestic Pigeons we have collected blood clots samples. These samples were tested and have been now added to the manuscript.
We have added in text at MM: “Additionally, from the 21 dead Columba livia domestica blood clots were collected (Table 2).” And added the new data to Table 2, Blood clots column.
We have added in text at Results: “All 21 blood clots samples collected from deceased Domestic Pigeons were successfully used for sex identification, and the results were identically with those obtained from oral swabs or feathers.”
AU: We have added in text at Discussions: “In some cases, the female-specific W-chromosome PCR band was consistently weaker than the Z-chromosome band, therefore, the weaker amplification of the diagnostic band could result in females being misdiagnosed as males [16]. In such cases, samples from females would show only a single band (from the W-chromosome), that could result in them being sexed as males, while samples from males would show absolutely no band [16,31]. For this reason, in the present study, all birds identified as male in only one sample type (oral swabs or feathers) were considered as unidentified. PCR female results are definitely reliable (two bands Z and W-chromosomes) while PCR male results (Z-chromosome) cannot be guaranteed as it could actually be a false negative for a female. On the other hand, the myth of the universal sexing marker has been debunked for Psittaciformes, because of different patterns of the PCR products depending on the tested species [31].”
Lines 186-196. Discuss/mention expected amplicon sizes.
AU: We assigned sex by counting the number of visible bands in each lane and estimating fragment lengths relative to the 100 bp DNA ladder.
We have changed in text at Results: “The PCR products showed two bands in females corresponding to CHD1W and CHD1Z genes and a single band in male corresponding to CHD1Z gene (Figure 2 and Figure 3), which range in size between 350 bp and 450 bp.”
Lines 207-280. Please comment on the following:
- I did not see that DNA quantities were measured. Perhaps this might have contributed to oral swabs having an statistical difference over feathers. If possible, please, test DNA using a nanodrop or similar technology. This can shed light on the differences between these samples. This will enrich the paper and make it more citable. In any case, please comment on the lack of this data.
AU: We have measured by Nanodrop the DNA samples obtained from oral swabs and feathers, but randomly. We have obtained DNA quantities between 16.5-103.3 ng/µl (A260/A280 ratio 2.34-1.97) in DNA samples obtained from oral swabs and between 3.5 -192.8 ng/µl (A260/A280 ratio 2.69-1.94) in DNA samples obtained from feathers. The quantity of DNA is variable depending on the number of cells collected on swab and the lesions of oral cavity, and the size of the calamus of the feathers.
In a previous research [25], we have measured by Nanodrop the DNA quantities in feathers processed by different technique. The highest DNA quantity was obtained by processing feathers samples with TissueLysser (Qiagen) before extraction.
- No comment is done over on Harvey et al 2006 findings, that some females can be scored as males by PCR. Please comment and explore this possibility. Add proper citations.
AU: We have added in text: “In some cases, the female-specific W-chromosome PCR band was consistently weaker than the Z-chromosome band, therefore, the weaker amplification of the diagnostic band could result in females being misdiagnosed as males [16]. During PCR annealing temperatures greater than 53°C for females and 58°C for males can product no bands from the Z-chromosome locus [16]. In such cases, samples from females would show only a single band (from the W-chromosome), that could result in them being sexed as males, while samples from males would show absolutely no band [16,31]. In current study, the PCR annealing temperatures were 48°C and no bands of different size than expected were observed. For these reasons, the possibility of some females being identified as false males due to poor amplification of diagnostic band was excluded, instead, all birds identified as male only in one type of samples (oral swab or feathers) were considered as unidentified. On the other hand, the myth of the universal sexing marker has been debunked for Psittaciformes, because of different patterns of the PCR products depending on the tested species [31].”
Lines 221-226. If possible, please comment on the presence of feather pulp on feathers collected.
AU: We have added in text: “Moreover, for higher DNA quantity, 2 to 4 contour feathers with intact calamuses sampled from the wings or abdominal region of birds were processed for each individual.”

Reviewer 2 Report
General comment
The study conducted is certainly interesting but there is a major flaw in the methodology of your experimental design. This implies to be very cautious with the interpretation of the results. Indeed, in the results (lines 186-187) you speak about the “overall PCR success rate of sex determination” which is formally correct as here you handle only about the fact that you see or not a PCR product (“a PCR success rate”). However, in the discussion afterwards, it becomes the ‘success rate of sex determination’ and this is a completely different concept because there you interpret the PCR results. The success rate of sex determination states if the sexing has been correctly performed. As such, with the data you have, it is impossible to gather these data as you don’t have any reference to decide whether the obtained PCR result is right or not. In the article [23] that you cited, the PCR is done in parallel on blood. Here you don’t have such data which means that it is always the same with sexing in birds: only the female trait in this case is discriminative. If you don’t see it, it is either a male or a false negative for a female (and you say yourself that the W-chromosome band is consistently weaker [line 65]). That is a major limitation of your study and the reader should be aware of it. I don’t say you cannot come finally to some conclusions rather close to the ones you had but you should stress that this depends on some hypotheses that you made and the reader has to be aware of these hypotheses.
I just can suggest a few hypotheses that you might formulate in order to be able to get at least some figures about the success rate of sex determination:
1) What is completely missing is if results from same individuals are comparable with the two methods (matching pairs of results on a same individual with the two methods). For instance for juvenile pigeons you get 6 males and 14 females as well with oral swabs as with feathers : are these individuals the same? I guess it is but it not explicitly indicated. It is important to identify the individuals for which the two tests come to the same sexing result. One may consider that if the results match with the two methods on a same individual, both methods provide the correct result (strictly this remains a hypothesis and it is probably not always true, if for some reason only small amounts of DNA can be collected in both methods, the male results will not always be reliable).
2) You should isolate from these common results obtained with hypothesis 1, the ones which don’t match with the two methods and separate them in : M, F and U. For instance in juvenile rose-ringed parakeets, I guess that with oral swab results for 8 male and 3 female results correspond to those obtained with feathers. This means you have 1 M and 3 F as specific results of the oral swabs. Out of these specific results, the female results are certainly reliable while the male ones cannot be guaranteed as it could be in fact a false negative result for a female. In the calculation you should discard these specific male results and add them to the undefined ones.
By combining the reliable figures linked to the above two hypotheses, you can come to a sex determination success rate for both tests but some male results will have to ranked in the category of unreliable results.
Specific comments
Line 28 : “…from feather [without final s] samples” (same also at line 208).
Line 66 : instead of “less amplification”, I would speak about “weaker amplification”
Line 88 : instead of speaking about “shorter”, I would state “faster”
Line 108 : “From” instead of “Form”
Figures 3 and 4 : why don’t you use the classical “male” symbol with an arrow at 45° above the horizon instead of an arrow that is perpendicular to the horizon?
Figure 3 : why do you color the bands of lines 1.4 and 1.6 with a red spot? It does not allow to see if it is a single bright band. In fact, for those lanes a new gel with less material loaded in the well would have been better because here it is impossible to discriminate if the bright band is a single W or single Z band or a combination of W and Z.
Line 265 : start sentence by “The current study included both …”
Author Response
Reviewer #2
The study conducted is certainly interesting but there is a major flaw in the methodology of your experimental design. This implies to be very cautious with the interpretation of the results. Indeed, in the results (lines 186-187) you speak about the “overall PCR success rate of sex determination” which is formally correct as here you handle only about the fact that you see or not a PCR product (“a PCR success rate”). However, in the discussion afterwards, it becomes the ‘success rate of sex determination’ and this is a completely different concept because there you interpret the PCR results. The success rate of sex determination states if the sexing has been correctly performed. As such, with the data you have, it is impossible to gather these data as you don’t have any reference to decide whether the obtained PCR result is right or not. In the article [23] that you cited, the PCR is done in parallel on blood. Here you don’t have such data which means that it is always the same with sexing in birds: only the female trait in this case is discriminative. If you don’t see it, it is either a male or a false negative for a female (and you say yourself that the W-chromosome band is consistently weaker [line 65]). That is a major limitation of your study and the reader should be aware of it. I don’t say you cannot come finally to some conclusions rather close to the ones you had but you should stress that this depends on some hypotheses that you made and the reader has to be aware of these hypotheses.
AU: We have changed in all text: “success rate of sex determination” with “PCR success rate of sex determination”
AU: From the 21 deceased Domestic Pigeons we have collected blood clots samples. These samples were tested and we have now added them to the manuscript.
We have added in text in MM: “Additionally, from the 21 dead Columba livia domestica blood clots were collected (Table 2).”
We have added in text in Results: “All 21 blood clots samples collected from deceased Domestic Pigeons were successfully used for sex identification, and the results were identically with those obtained from oral swabs or feathers.”
I just can suggest a few hypotheses that you might formulate in order to be able to get at least some figures about the success rate of sex determination:
1) What is completely missing is if results from same individuals are comparable with the two methods (matching pairs of results on a same individual with the two methods). For instance for juvenile pigeons you get 6 males and 14 females as well with oral swabs as with feathers : are these individuals the same? I guess it is but it not explicitly indicated. It is important to identify the individuals for which the two tests come to the same sexing result. One may consider that if the results match with the two methods on a same individual, both methods provide the correct result (strictly this remains a hypothesis and it is probably not always true, if for some reason only small amounts of DNA can be collected in both methods, the male results will not always be reliable).
AU: We have added in Abstract: “The similarity between the oral swab and feathers was obtained in 78.38% (58/74; CI 95%: 67.73-86.23) birds.”
We have added in text at MM: “From December 2021 to March 2022 samples of oral swabs and feathers were randomly collected from 101 companion birds from the orders Columbiformes and Psittaciformes (Columba livia domestica, Psittacula krameri, Neophema splendida and Agapornis spp.) in Romania. The study included 31 adult birds and 70 juveniles [2,4], of which 80 were live birds belonging to all four species tested and 21 cadavers belonging to Columba livia domestica. From these 101 birds, 74 paired samples of feathers-oral swabs from the same individuals and 27 additional oral swabs from featherless newly hatched birds were collected [2,4] (Table 2). Additionally, from the 21 dead Columba livia domestica blood clots were collected (Table 2).”
We have added in text at Results: “Conventional PCR with P2 and P8 primers allowed sex determination in all four tested bird species. The results of molecular sexing of birds from oral swabs and feath-ers are presented in Table 3. All birds identified as male only in one type of samples (oral swab or feathers) were considered as unidentified.”
We have added in text at Results: “A similarity between the sex determination from the oral swab versus feather samples was obtained in 58/74 birds (78.38%; CI 95%: 67.73-86.23). In Domestic Pigeons the same results from oral swabs and from feathers was obtained in 38/43 birds (88.37%; CI 95%: 75.52-94.93), 5 samples from different dead pigeons being unidentified (3 from oral swabs collected from dead pigeons that had lesions in the oral cavity and 2 from feathers). All 21 blood clots samples collected from deceased Domestic Pigeons were successfully used for sex identification, and the results were identically with those obtained from oral swabs or feathers. In Rose-ringed Parakeets the similarity between oral swab and feathers was 73.33% (11/15; CI 95%: 48.05-89.10), in Love-birds 87.50% (7/8; CI 95%: 52.91-97.76) and in Scarlet-chested Parrot 25% (2/8; CI 95%: 7.15-59.07). In one individual of Lovebirds the sex could not be identified because neither the oral swab nor the feather samples could be amplified.”
2) You should isolate from these common results obtained with hypothesis 1, the ones which don’t match with the two methods and separate them in : M, F and U. For instance in juvenile rose-ringed parakeets, I guess that with oral swab results for 8 male and 3 female results correspond to those obtained with feathers. This means you have 1 M and 3 F as specific results of the oral swabs. Out of these specific results, the female results are certainly reliable while the male ones cannot be guaranteed as it could be in fact a false negative result for a female. In the calculation you should discard these specific male results and add them to the undefined ones.
By combining the reliable figures linked to the above two hypotheses, you can come to a sex determination success rate for both tests but some male results will have to ranked in the category of unreliable results.
AU: From the 21 deceased Domestic Pigeons we have collected blood clots samples. These samples were tested and we have now added them to the manuscript.
We have added in text in Discussion: “All blood clots samples collected from deceased Domestic Pigeons were successfully used for sex identification, and the results were identically with those obtained from oral swabs or feathers.”
AU: We have added in text at Results: “All birds identified as male only in one type of samples (oral swab or feathers) were considered as unidentified.”
We have modified the data in Table 2, and we recalculated all the percentages from the Results. “Legend: M – male; F – female; U – unidentified; PCR Success rate - percentage of samples whose sex was successfully identified using oral swabs and feather samples; * - not applicable; ** - birds identified as male only in one type of samples (oral swab).”
AU: We have added in text: “In some cases, the female-specific W-chromosome PCR band was consistently weaker than the Z-chromosome band, therefore, the weaker amplification of the diagnostic band could result in females being misdiagnosed as males [16]. During PCR annealing temperatures greater than 53°C for females and 58°C for males can product no bands from the Z-chromosome locus [16]. In such cases, samples from females would show only a single band (from the W-chromosome), that could result in them being sexed as males, while samples from males would show absolutely no band [16,31]. In current study, the PCR annealing temperatures were 48°C and no bands of different size than expected were observed. For these reasons, the possibility of some females being identified as false males due to poor amplification of diagnostic band was excluded, instead, all birds identified as male only in one type of samples (oral swab or feathers) were considered as unidentified. On the other hand, the myth of the universal sexing marker has been debunked for Psittaciformes, because of different patterns of the PCR products depending on the tested species [31].”
We have added a new reference: “31. Kroczak, A.; Wołoszyńska, M.; Wierzbicki, H.; Kurkowski, M.; Grabowski, K. A.; Piasecki, T.; Urantówka, A. D. New Bird sexing strategy developed in the order Psittaciformes involves multiple markers to avoid sex misidentification: Debunked myth of the Universal DNA marker. Genes 2021, 12(6), 878, doi: 10.3390/genes12060878.”
Specific comments
Line 28 : “…from feather [without final s] samples” (same also at line 208).
AU: Done.
Line 66 : instead of “less amplification”, I would speak about “weaker amplification”
AU: Done.
Line 88 : instead of speaking about “shorter”, I would state “faster”
AU: Done.
Line 108 : “From” instead of “Form”
AU: Done.
Figures 3 and 4 : why don’t you use the classical “male” symbol with an arrow at 45° above the horizon instead of an arrow that is perpendicular to the horizon?
AU: Done. We have changed Figures 2 and 3 and their legends.
Figure 3: why do you color the bands of lines 1.4 and 1.6 with a red spot? It does not allow to see if it is a single bright band. In fact, for those lanes a new gel with less material loaded in the well would have been better because here it is impossible to discriminate if the bright band is a single W or single Z band or a combination of W and Z.
AU: The red spots indicate strong PCR amplification with a high yield of DNA. We have changed Figures 2 and 3 and their legends.
Line 265 : start sentence by “The current study included both …”.
AU: Done.

Reviewer 3 Report
The paper is clearly and concisely written. The literature used in the introduction and discussion is adequate, although not the most relevant in this area. The methodology is appropriate. The results are clearly presented in both text and images.
However, the main problem is related to scientific significance. Namely, the work deals with a current topic more than a decade ago. In that period, commercial kits for DNA isolation could not efficiently deal with PCR reaction inhibitors (keratin in feathers, haemoglobin in the blood, etc.) and this topic was very important for everyone who dealt with this issue due to a significant number of failed reactions.
With the advancement of technology and the appearance of new modern nucleic acid isolation kits, this problem has been overcome. It is no longer even close to the focus of the scientific community. In my lab, there is no known (certainly in the last 10 years) failure to isolate nucleic acid when using samples of avian origin regardless of which sample is used as the DNA source (even if FFPET is used). Therefore, a paper with this title (and this content) would certainly not be interesting to read.
Author Response
Reviewer #3
The paper is clearly and concisely written. The literature used in the introduction and discussion is adequate, although not the most relevant in this area. The methodology is appropriate. The results are clearly presented in both text and images.
However, the main problem is related to scientific significance. Namely, the work deals with a current topic more than a decade ago. In that period, commercial kits for DNA isolation could not efficiently deal with PCR reaction inhibitors (keratin in feathers, hemoglobin in the blood, etc.) and this topic was very important for everyone who dealt with this issue due to a significant number of failed reactions.
With the advancement of technology and the appearance of new modern nucleic acid isolation kits, this problem has been overcome. It is no longer even close to the focus of the scientific community. In my lab, there is no known (certainly in the last 10 years) failure to isolate nucleic acid when using samples of avian origin regardless of which sample is used as the DNA source (even if FFPET is used). Therefore, a paper with this title (and this content) would certainly not be interesting to read.
AU: Feathers-versus oral swabs-based sex determinations were previously studied [17,22,25,27,31], but not in the species mentioned in the current study.
We have added in Introduction: “Moreover, from our knowledge, the current study reports for the first-time feathers-versus oral swabs-based sex determinations in the species Psittacula krameri, Neophema splendida and Agapornis spp.”
AU: Moreover, in current study we used our improved method of feather processing (TissueLyser with steel beads), in order to obtain a larger amount of DNA.
AU: Considering that blood and feathers have been commonly used as a source of DNA for molecular sexing in birds, we demonstrated that oral swabs can be a reliable approach to obtain good quality DNA. Oral swabs can be used successfully as a source of DNA for newly hatched featherless birds. Oral swabs as feathers, as sources of DNA have limitations, ex. the lesions of the oral cavity etc.
AU: We have added in text at Results: “Oral swabs proved to be a more reliable sample for genetic sex determination in the species tested in this study.”
AU: We have added in text at Conclusions: “The present study includes the comparison between paired samples of feathers-oral swabs in Psittacula krameri, Neophema splendida, Agapornis spp. and Columba livia domestica. The oral swab samples proved to be more efficient compared to feather samples and provided adequate DNA templates for molecular sexing of juvenile and adult companion birds. Oral swab samples were exclusively used for newly hatched featherless parrots. Current study recorded a limitation of oral swab samples in the case of the presence of the pseudo-membrane deposits in the oral cavity. Further improvements of PCR protocols for noninvasive and moderate invasive sampling are needed in order to eliminate the invasive methods in birds.”

Round 2
Reviewer 2 Report
The manuscript is improved with the changes that were made.
Going again through the text, I have three editorial remarks:
1° I would add a small further explanation at the end of the first paragraph of results (thus extending line 168):
... were considered as unidentified, [except if the result could be confirmed by another independent analysis (PCR on blood clots).] - The addition is given between the brackets.
2° In line 188, it should be 'identical' and not 'identically'.
3° There is a redundancy in the discussion : the content from line 257 (starting from "No results...") to line 262 is repeated in lines 263-270 (up to cadavers [17,19,28,29,30]). I would suggest to skip the lines 257-262.
Author Response
Reviewer #2
The manuscript is improved with the changes that were made.
Going again through the text, I have three editorial remarks:
1° I would add a small further explanation at the end of the first paragraph of results (thus extending line 168):... were considered as unidentified, [except if the result could be confirmed by another independent analysis (PCR on blood clots).] - The addition is given between the brackets.
AU: Done. We have added in Results: “except if the result could be confirmed by another independent analysis (PCR on blood clots)”.
2° In line 188, it should be 'identical' and not 'identically'.
AU: Done.
3° There is a redundancy in the discussion : the content from line 257 (starting from "No results...") to line 262 is repeated in lines 263-270 (up to cadavers [17,19,28,29,30]). I would suggest to skip the lines 257-262.
AU: Done.
AU: Thank you for the constructive criticisms, useful comments and thoughtful suggestions!

Reviewer 3 Report
I am grateful for the answers and the changes made
Author Response
Reviewer #3
I am grateful for the answers and the changes made.
AU: We have changed in Introduction “The aim of this study was to compare the results obtained by PCR performed on isolate genomic DNA from paired samples of feathers (processed using TissueLyser (Qiagen) in order to increase the amount of DNA) and oral swabs, collected from the same individuals. It is also aimed to find an alternative to invasive sampling (blood collection) for sexing newly hatched birds. The present study offers oral swab sampling as a moderately invasive solution for sexing featherless newly hatched chicks. According to our knowledge, the current study reports for the first-time feathers-versus oral swabs-based sex determinations in the following species, Psittacula krameri, Neophema splendida and Agapornis spp. Pairing parrots has been proven to increase welfare, therefore early sex determination could present great value for bird owners.”
AU: We have added in text in Discussions: „If a decade ago commercial kits for DNA isolation could not effectively deal with inhibitors of the PCR reaction (keratin in feathers, hemoglobin in blood, etc.) and a significant number of failed reactions were recorded, with the advancement of technology and the appearance of new modern kits of nucleic acid isolation, this problem has been overcome. However, even if this problem has been solved, genetic sex determination still has limitations such as: the size of feather samples, the degree of feather damage with or without intact calamuses, the anatomical region from where the feathers have been plucked, the number of cells collected on oral swabs or lesions in the oral cavity.”
